# Novel Insights into Fungal Infections Prophylaxis and Treatment in Pediatric Patients with Cancer

**DOI:** 10.3390/antibiotics11101316

**Published:** 2022-09-27

**Authors:** Gianluca Bossù, Riccardo Di Sario, Edoardo Muratore, Davide Leardini, Andrea Pession, Susanna Esposito, Riccardo Masetti

**Affiliations:** 1Department of Medicine and Surgery, Pediatric Clinic, University Hospital, University of Parma, 43126 Parma, Italy; 2Pediatric Unit, IRCCS Azienda Ospedaliero-Universitaria di Bologna, 40138 Bologna, Italy

**Keywords:** antifungal stewardship, antifungal drugs, fungal infection, invasive fungal disease, pediatric cancer

## Abstract

Invasive fungal diseases (IFDs) are a relevant cause of morbidity and mortality in children with cancer. Their correct prevention and management impact patients’ outcomes. The aim of this review is to highlight the rationale and novel insights into antifungal prophylaxis and treatment in pediatric patients with oncological and hematological diseases. The literature analysis showed that IFDs represent a minority of cases in comparison to bacterial and viral infections, but their impact might be far more serious, especially when prolonged antifungal therapy or invasive surgical treatments are required to eradicate colonization. A personalized approach is recommended since pediatric patients with cancer often present with different complications and require tailored therapy. Moreover, while the *Aspergillus* infection rate does not seem to increase, in the near future, new therapeutic recommendations should be required in light of new epidemiological data on Candidemia due to resistant species. Finally, further studies on CAR-T treatment and other immunotherapies are needed in patients with unique needs and the risk of complications. Definitive guidelines on IFD treatment considering the evolving epidemiology of antifungal resistance, new therapeutic approaches in pediatric cancer, novel antifungal drugs and the importance of an appropriate antifungal stewardship are urgently needed.

## 1. Introduction

Invasive fungal diseases (IFDs) are a relevant cause of morbidity and mortality in children with cancer [1]. Several factors are responsible for a higher susceptibility to developing IFD in these patients, such as prolonged neutropenia, steroid exposure, central line catheters and graft-versus-host disease (GvHD) in those receiving hematopoietic stem cell transplantation (HSCT) [1]. Diagnosis of IFD often represents a diagnostic conundrum because in some cases fever can be the only manifestation and early recognition is challenging [2,3,4]. These patients are indeed eligible for prophylaxis with both antifungal compounds and non-medical interventions, such as the use of air filtration, granting a better outcome [5]. Early treatment of IFD is fundamental to reducing mortality. Empirical antifungal therapy is necessary when the clinical suspicion of IFD is high. Moreover, the correct management of IFD can prevent possible delays in treatment, impacting patients’ outcomes and the changing epidemiology of IFD should be appropriately considered [6]. The 8th European Conference on Infections in Leukemia (ECIL-8) recently updated its guidelines regarding the diagnosis and management of IFD in pediatric patients with cancer or post-HSCT [7]. The aim of this review is to highlight the rationale and novel insights into antifungal prophylaxis and treatment in pediatric patients (between birth and 18 years) with oncological and hematological diseases. In the absence of pediatric data and recommendations, the results extrapolated from adult cohorts are presented.

## 2. Incidence, Risk Factors and Epidemiology

The incidence of IFDs in pediatric cancer patients can vary with the underlying pathology, ranging from 3.7 to 28.0% in acute myeloid leukemia (AML) [8,9], from 0.6 to 11.0% in newly diagnosed acute lymphoblastic leukemia (ALL), from 4 to 6% in relapsed ALL and from 8 to 21% in severe aplastic anemia (SAA) [10,11,12,13]. AML has historically been considered to be associated with a higher risk of IFD than ALL, since myeloid cells are critical in the inhibition of fungal proliferation and neutropenia is more persistent after chemotherapy for AML [14,15]. Regarding solid tumors, the risk for IFD is generally considered lower: 0.4% for lymphomas and 0.1% for other solid tumors [16]. The ECIL-8 guidelines maintain a precise stratification of risk, depending on the kind of pathology [7], identifying high-risk patients affected by AML, relapsed ALL, or undergoing allogeneic HSCT and medium-risk children affected by ALL, non-Hodgkin lymphoma and those receiving autologous HSCT. Sporadic occurrence is for pediatric solid tumors, brain tumors and Hodgkin lymphoma. These differences can be explained by the different risk factors for IFDs. Oncological blood disorders are the main risk factor due to inefficient immune surveillance, followed by severe and prolonged neutropenia due to intensive chemotherapy [11,13]. Prolonged use of corticosteroids at a dosage higher than 0.3 mg/kg/day, transplantation of both stem cells and lungs, and GvHD are other important risk factors [11,13].

For epidemiology, historically, *C. albicans* has been the most relevant cause of IFD in children undergoing chemotherapy for cancer or HSCT and fluconazole was the leading molecule for antifungal prophylaxis [6]. The widespread use of fluconazole as a primary antifungal prophylaxis granted a progressive decrease in *C. albicans* infections and favored a progressive increase in the incidence of IFDs caused by *non-albicans-Candida* species, which are resistant to drugs commonly used for treatment of candidiasis, filamentous fungi, multidrug resistant (MDR) fungi and molds [6]. Even though *C. albicans* is no longer the leading cause of IFDs in immunocompromised children, it is still a relevant cause of morbidity and mortality, as noted in a study by Yeh et al. that documented 29 episodes of fungal infections in 26 children in the period between January 1987 and October 2005, with *Candida* species being the leading pathogens (14/29, 40%), including *C. albicans*, *C. tropicalis* and *C. parasilosis*. The cited study is an example of the trending epidemiology with a progressive shift to non-albicans species causing IFDs, among which, alongside the already cited *C. tropicalis* and *C. parasilosis*, we can count *C. krusei*, *C. glabrata* and *C. lusitanii* [17,18]. This is not only epidemiologic information because some of the non-albicans species are resistant to drugs commonly used for the treatment of IFD (i.e., *C. krusei*, *C. glabrata*, and recently reported *C. auris*) [18,19,20]. *C. krusei* has shown intrinsic resistance to fluconazole [18,20]; *C. glabrata* can account for a high resistance rate to echinocandin and even MDR species are an emerging threat [21]; *C. auris* is another merging non-*albicans* species with MDR features (in particular, resistance to azoles, polyenes and echinocandins) [22,23]. The treatment of such resistant fungi is challenging and the choice of the molecule should be patient-tailored and based on practice and sensitivity assays, choosing broad-spectrum antifungal drugs, better if newly discovered and less used, such as isavuconazole.

The shifting of the epidemiological landscape of IFDs in such patients leads to *Aspergillus* spp. becoming the most frequent cause of fungal infections [6]. Nonetheless, as for non-*albicans* spp., non-*fumigatus* species are emerging and the most relevant are *A. niger*, *A. terreus* and *A. nidulans* [11,24]. Moreover, as for *Candida* spp., drug resistance is a huge threat and a complex reality to face: some *A. fumigatus* species are showing azole resistance interestingly mediated by mutations in a specific gene (*TR34/L98H*) and could have occurred due to the vast use of triazole fungicides in agriculture [25]. Non-*fumigatus* spp. to have an increasingly drug resistance rate, such aas some *A. terreus* specimens resistant to amphotericin B and MDR samples, are emerging even [26].

Alongside these considerations, in a recent review of invasive IFD, Pana et al. noted an increasing shift toward non-*Aspergillus* molds (NAM) [27]. NAM constitutes 10–25% of all invasive mold diseases in patients with hematological malignancy or post-HSCT [28]. The most common NAMs documented in the literature, taking some data from adults, are *Mucorales*, i.e., opportunistic and ubiquitous fungi causing Mucormycosis, affecting bloodstream with potential dissemination to every organ [29]. Liposomal amphotericin B is the main therapy for its in vitro susceptibility, established central nervous system (CNS) penetration, and favorable side-effect profile [28]. Another NAM is *Scedosporium* spp., with *S. apiospermum* and *S. aurantiacum* as the most common causative agents of *Scedosporiosis*, which can interest joints, bones, CNS or low airways, evolving in fatal disseminated disease. The cornerstone of treatment is currently i.v. voriconazole, as for adults, is considered a surgical debridement in case of a localized site of infection [28,30]. *Lomentospora prolificans* is another emerging NAM that is clinically similar to Scedosporiosis, but there are few data in pediatrics [31]. The last emerging NAM is represented by *Fusarium* spp., causing fusariosis, mostly present among Australian studies. Clinically beginning with cutaneous manifestations, it can cause endophthalmitis, sinusitis, meningitis, pneumonia and bloodstream infection. The empiric first-line therapy can be managed with liposomal amphotericin B, posaconazole or voriconazole [28,32].

The last molds to be mentioned, even if rarer than those cited before, are some belonging to Hyalohyphomycoses: *Paecilomyces variotii*, a saprophytic mold that can cause a broad spectrum of diseases (pneumonia, skin and soft tissue infections, osteomyelitis, peritonitis and fungaemia), and *Purpureocillium lilacinum* (formerly *Paecilomyces lilacinum*), affecting mostly skin and eye (keratitis and endophtalmitis); both are rare and rationale for treatment needs in vitro sensitivity assays even though an empiric therapy with liposomal amphotericin B or posaconazole can be started [28,33,34].

## 3. Rationale for Fungal Infection Prophylaxis and Treatment in Acute Leukemia and Hematopoietic Stem Cell Transplantation Recipients

IFDs are usually very difficult to eradicate in immunocompromised children and consequently their prevention has always been of great importance, both with pharmacological and “non-medical” interventions, including modernization of air filters [35,36]. Regarding antifungal prophylaxis, two recent randomized studies should be mentioned. Fisher et al. compared the efficacy of prophylaxis with capsofungin or fluconazole in 517 pediatric patients affected by AML [37]. The 5-month cumulative incidence of proven or probable IFD was 3.1% (95% confidence interval [CI], 1.3–7.0%) in the caspofungin group vs. 7.2% (95% CI, 4.4–11.8%) in the fluconazole group (overall *p* = 0 .03 by log-rank test) and no significant differences in 2-year overall survival (68.8% caspofungin vs. 70.8% fluconazole, overall *p* = 0.66 by log-rank test) were observed. However, this study was terminated early due to an unplanned interim analysis that suggested futility [37]. Recently, Dvorak et al. analyzed antifungal prophylaxis in 560 children undergoing HSCT comparing caspofungin and a triazole, fluconazole or voriconazole [38]. The cumulative incidence of proven or probable IFD at day 42 was 1.4% (95% CI, 0.3–5.4%) in the caspofungin group and 1.4% (95% CI, 0.4–5.5%) in the triazole group (*p* = 0.99). Consequently, the authors stated that prophylaxis with caspofungin did not significantly reduce the cumulative incidence of IFD, but in this case, the trial was closed early [38]. On this matter, it is also important to consider the recent guidelines that have been independently elaborated by the Children’s’ Oncology Group (COG) [39] and the ECIL-8 [13], taking into account all evidence available at the moment. Their main differences and recommendations are highlighted in Table 1.

Regarding the treatment of IFD, the duration of therapy for candidemia is 14 days after negative blood cultures and neutrophil recovery, whereas it is not defined for aspergillosis. Because of the increase in *C. krusei* and *C. glabrata*, resistance testing is fundamental in candidemia and should direct the therapeutic approach. In all patients with IFD, thorough examination of all possible sites of infection (especially the central nervous system for aspergillosis) is always required [13]. The management of Mucormycosis is similar to Aspergillosis and centered on the prompt initiation of therapy with amphotericin B and surgery [6,40,41,42]. Historically burdened by difficult treatment and heavy mortality, recent evidence has shown that implementation of isavuconazonium sulfate (i.e., a new broad spectrum triazole) in a multi—modal approach can eradicate Mucormycosis in patients undergoing HSCT [43]. Data extrapolated from adult cohorts suggest a loading dose of 10 mg/kg every 8 h for 6 doses, followed by 10 mg/kg dosing every 24 h (even if pediatric level goals are not yet established), a dosage that seems to be well tolerated. Finally, there are no available data regarding the pediatric population on empirical antifungal therapy, but it is reasonable in patients with 96 h of persistent/recurrent fever, granulocytopenia and unresponsive to broad-spectrum antibiotics. The rationale for empirical therapy derives from three randomized studies [44,45,46], according to which there are no differences in terms of efficacy between caspofungin and liposomal amphotericin; efficacy and safety appeared to be in line with evidence from larger adult cohorts. Itraconazole, voriconazole and micafungin are not approved for empirical therapy. Consequently, the switch to another antifungal drug seems reasonable in patients already receiving prophylaxis when empirical therapy must be initiated, even if there is no evidence in the literature regarding this matter. Table 2 reports a summary of the main studies regarding the prophylaxis and treatment of IFD.

## 4. Breakthrough Invasive Fungal Infections

The widespread use of antifungal prophylaxis is causing a shift in the epidemiology of fungal infections as a direct consequence, but a new emerging reality is represented by breakthrough invasive fungal infections (IFI) in children, assuming azoles or echinocandins as prophylaxis [54]. The correct definition of breakthrough IFIs is not still established and shared worldwide, but in 2019, the Mycoses Study Group Education and Research Consortium (MSG-ERC) and the European Confederation of Medical Mycology (ECMM) defined it as any IFI occurring during exposure to antifungal drugs, including fungi outside the spectrum of activity of an antifungal, with the period depending on pharmacokinetic properties and extending at least until one dosing interval after drug discontinuation. The time of breakthrough IFI was defined as the first attributable clinical sign or symptom, mycological finding or radiological feature [55]. Posaconazole is probably the most used drug for prophylaxis and Lerolle et al. reported 5.3% of breakthrough infections in children undergoing HSCT, mainly due to *Aspergillus* spp. (44%) and *Candida* (25%) [56]. Even voriconazole prophylaxis in the HSCT setting was evaluated, with Wingard et al. reporting a 13% incidence in a 1-year study, mostly caused by *Aspergillus* [57]. Prophylaxis with echinocandins is also affected by cases of breakthrough infections: in a study conducted by Funaki et al., 8 cases of breakthrough candidemia were reported in children with hemato-oncologic malignancies under prophylaxis with micafungin; *C. parapsilosis* and *C. albicans* were isolated [58]. This may be due either to clinical resistance to echinocandins or to the presence of reservoirs of infections, such as central catheters, favoring the formation of biofilms. Additionally, a five-year retrospective review reported cases of breakthrough filamentous mold infection while on caspofungin therapy [59]. These children were affected by ALL, AML, Burkitt’s lymphoma and SAA and six confirmed positive cultures (*A. fumigatus*, *A. niger*, *Fusarium oxysporum*, *Alternaria infectoria* and *Rhizomucor pusillus*) were reported with *Fusarium oxysporum* and *R. pusillus*, considered inherently resistant to caspofungin, maybe due to *FKS1* mutations [59]. However, few data are available for this phenomenon in pediatrics, and further studies are necessary for its correct definition and to establish guidelines for diagnosis and treatment.

## 5. Feasibility of Hematopoietic Stem Cell Transplantation in Patients with Invasive Fungal Disease

IFDs are a frequent and deadly complication in HSCT receivers, with a higher incidence rate in haploidentical and mismatched unrelated donor HSCT than in autologous HSCT [60]. This difference depends on the longer duration of neutropenia in the former and the presence of GvHD. Both acute and chronic GvHD appear to be relevant risk factors for IFD, with acute GvHD alone increasing by 2.4–5.7 the hazard ratio for invasive Aspergillosis [61,62,63]. It is still unclear whether this effect is independent or strictly linked to the high-dose corticosteroid regimens used to treat GvHD [61,64]. Figure 1 summarizes timing, risk factors, epidemiology and prophylaxis indications for IFD in the context of HSCT.

For this increased risk, the presence of an IFD has long been considered a contraindication to HSCT, but thanks to the advancement in early diagnosis and therapy, it has become more common and feasible nowadays [65], although the evidence on this matter is still limited [66,67]. Regarding pediatric patients, a recent case series described the characteristics of 23 children with prior or active IFD undergoing allogenic HSCT [68]. Fourteen patients had a recent diagnosis of IFD, in most cases aspergillosis, and were still on secondary prophylaxis with liposomal amphotericin B or an azole agent when they underwent HSCT, while the other 9 were still treated for active disease at the start of the conditioning regimen and 4 also received granulocyte transfusion. Four of these nine children with active IFD developed a new one post-transplant (n = 2) or experienced progression of their original IFD (n = 2). Acute GVHD occurred in 9 patients and four had severe GVHD; 3 patients developed chronic GVHD. One year after HSCT, 5 children died, one of them for direct consequences of the underlying IFD (uncontrolled pulmonary aspergillosis and GVHD). The authors concluded that most patients did not develop a new IFD after HSCT and did not evidence an increase in mortality, so, according to their opinion, a recent history of IFD should not be a contraindication for HSCT in pediatric patients [68]. However, the scarcity of pediatric data imposes further caution, as seen in adult patients [69], and the implementation of new strategies.

## 6. Granulocyte Transfusion and Invasive Fungal Disease

Granulocyte transfusion (GTX), a medical procedure in which granulocytes are infused in patients with neutropenia secondary to chemotherapy or transplantation, might represent a tool in the case of pre-existent IFD [70,71]. Definitive guidelines are not yet available, and the reduction of incidence in IFD in patients receiving GTX is not fully supported by strong evidence, although it should be noted that, generally, these patients are already critically ill [70,71]. An ongoing multi-center trial on this matter will possibly provide more helpful results [72].

Accurate selection of patients eligible for GTX is probably the key to its success, with the efficacy of granulocyte transfusions deriving mostly from the cell dose per body weight and time [73]. Technological improvement in the collection of red blood cells from donors with more modern sedimentation agents will hopefully make transfusions safer [74,75]. An interesting approach for the implementation of GTX is to use it as a bridge to transplant if treatment of IFD is still incomplete at the time of HSCT, or to bridge a secondary post-HSCT period of neutropenia after graft failure [68]. GTXs are infused after day 0 and continued thrice a week for generally 3–4 weeks until donor engraftment is achieved [76,77,78].

The pediatric cohorts found in the literature are quite limited, as summarized in Table 3 [77,78,79,80,81].

## 7. Fungal Infections in Children Receiving CAR-T Cell Therapy

CAR-T cell therapy is a type of treatment in which a patient’s T cells are changed in the laboratory so they will attack cancer cells [82]. T cells are taken from a patient’s blood. Then, the gene for a special receptor that binds to a certain protein on the patient’s cancer cells is added to the T cells in the laboratory. The special receptor is called a chimeric antigen receptor (CAR). Large numbers of CAR-T cells are grown in the laboratory and given to the patient by infusion. CAR T cells are able to bind to an antigen on cancer cells and kill them [82].

European Medicines Agency (EMA) has approved the use of tisagenlecleucel for the treatment of refractory/relapsed acute lymphoblastic leukemia (B-ALL) in children and young adults as well as relapsed/refractory diffuse large B-cell lymphoma [83]. The increased risk of infection is important during CAR-T treatment due to different complications in the immune response [84,85]. These complications include neutropenia, which is present in up to 80% of patients within the first month after the infusion of CAR T cells [83,86] and variable B cell aplasia with secondary hypogammaglobulinemia [87]. Cytokine release syndrome (CRS) and neurotoxicity are frequent side effects of CAR-T cell infusion: their treatment requires the use of corticosteroids and/or tocilizumab, i.e., drugs that might impair the immune response [88,89,90].

The infectious complications of CAR-T cell therapy have not yet been extensively studied in children. There is only one pediatric cohort study regarding infectious complications following CAR-T cell infusion in children, adolescents and young adults with relapsed/refractory B-ALL [91]. In this study, the authors reported that 33 of the 83 patients who were taken into consideration experienced one or more infections in the first 28 days after infusion. Of these 33 patients, the majority had bacterial infections, whereas only one patient was diagnosed with an invasive mold infection. It is interesting to note that this patient was already receiving voriconazole for a previously diagnosed invasive mold infection and had concomitant severe neutropenia with mild CRS treated with tocilizumab. The authors stated that it is unclear whether this infection represented new fungal disease or worsening of the previously diagnosed infection [91].

If we look at data extrapolated from adult cohorts, overall fungal infections in patients receiving CAR-T cell therapy ranged from 3 to 8%, thus representing a minority of the complications [86,89,90]. Pulmonary aspergillosis and candidemia are the most common IFD in these patients; their clinical manifestations were serious and usually occurred in patients heavily immunocompromised with prolonged neutropenia [86,89,90].

Prophylaxis for IFD in children treated with CAR-T therapy is strongly recommended. According to an electronic survey carried out by the American Society for Blood and Marrow Transplantation Pharmacy Special Interest Group involving 52 institutions, 87% of the respondents used antifungal prophylaxis, which in most cases had been fluconazole (92%) and generally discontinued neutrophil recovery (62%) [91,92].

Recently, the European Society for Blood and Marrow Transplantation (EBMT) and the European Hematology Association (EHA) published a series of recommendations on the management of adults and children receiving CAR-T cell therapy. The authors stated that IFI was rare and defined a common guideline for IFD prophylaxis, identifying posaconazole, fluconazole and micafungin as the drug of choice [93].

Evidence regarding the long-term complications and outcomes of IFD in pediatric patients treated with CAR-T is still insufficient. The wider implementation of CAR-T will probably optimize IFD management in each individual group of patients.

## 8. Fungal Infections and Immunotherapy

Antibody-based immunotherapies targeting B-cell surface markers have shown remarkable clinical activity in patients with hematological malignancies [94,95,96]. Blinatumomab and inotuzumab ozogamicin (InO) represent a new tool for the treatment of relapsed/refractory acute lymphoblastic leukemia [94,95,96]. While the adverse effects of immunotherapy, such as neurotoxicity or cytokine release syndrome, are well described [97], definitive data on infectious complications are still awaited. In relation to blinatumomab, a moderate risk for IFDs after treatments might be present, but similarly to CAR-T, it is still unclear if it is related to corticosteroids/tocilizumab use for neurotoxicity or the mechanism of the drug [98]. In the phase 3 trial TOWER 405, adult patients were randomized to receive chemotherapy or blinatumomab: while the general infection rate was lower in the blinatumomab group (34.1% vs. 52.3%), the number of mold-invasive infections was higher in the blinatumomab group [99]. Preliminary evidence indicates that infection rates in children receiving blinatumomab might be lower than in those treated with chemotherapy. In a randomized clinical trial that included 208 pediatric patients with high- and intermediate-risk first relapse of B-cell ALL, the cumulative rates of infection were 27% in the blinatumomab group and 70% in the chemotherapy group [100]. In contrast, Locatelli et al. reported a greater number of grade 3 infections in the blinatumomab group than in the chemotherapy cohort (18.5% vs. 9.8%, respectively) [101]. The authors justified these results, stating that the observational period ended later for the blinatumomab group, thus overlapping with other anticancer therapies [101]. In a retrospective study analyzing a cohort of 51 heavily pretreated children with relapsed/refractory ALL receiving InO, the authors reported 15 (29%) infectious toxicities describing two IFIs, one candidemia and one probable fungal infection [102]. Due to the scarcity of data on IFDs, it is still impossible to determine a unified approach for the management and/or prevention of children receiving targeted immunotherapies like blinatumomab or InO. Moreover, some authors have suggested that this lack of data imposes further caution when administering azole antifungals to avoid drug-to-drug interactions and adverse effects (for instance, InO increases the QTc, so heart rate monitoring is recommended with concomitant azole therapy) [103].

Blinatumomab and/or InO might act as “bridge therapy” not only for heavily pre-treated patients but also for patients with severe infectious complications awaiting subsequent more intense chemotherapy or HSCT. Contreras et al. reported the case of a 23-month-old patient B ALL that developed invasive Mucormycosis of the sinuses: the patient received one cycle of blinatumomab during active medical and surgical management of the fungal infection, tolerating the treatment without toxicity, and was later able to resume the post-induction therapy as originally planned [104]. More recently, a report from the Israeli Study Group of Childhood Leukemia described the use of Blinatumomab on 4 children with IFD (2 candidemia, 1 disseminated Mucormycosis and 1 rhinocerebral Mucormycosis plus pulmonary aspergillosis) [105]. Blinatumomab was administered to avoid therapy delay during IFD treatment and the patients subsequently resumed their planned treatment (one of them died of transplant-related complications). The authors also noted that, during the administration of blinatumomab, no new fatal or life-threatening complications were reported [105]. While experience is limited, these short courses of immunotherapy could be a new approach to avoiding the possible absence of treatment in pediatric patients burdened with IFD.

## 9. Novel Drugs for Contrasting Invasive Fungal Diseases

The growing prevalence and the high burden, costs and health-related consequences of IFD in immunocompromised patients has urged the need for novel antifungal drugs, some of which have shown a possible application in pediatric patients.

### 9.1. Isavuconazole

One of the most recent triazoles, isavuconazole, was approved by EMA for the treatment of Mucormycosis in 2015, and since then, it has gained increasing interest and has been used to treat a variety of fungal infections [106].

Data regarding the efficacy and long-term safety of isavuconazole in the pediatric population are scarce but encouraging. Isavuconazole at the studied dose of 10 mg/kg was generally well tolerated and showed an overall safety profile similar to that observed in adults; moreover, plasma drug exposure was comparable to that observed in adults in phase 3 efficacy trials [107]. More recent data [108] suggested that the variability of isavuconazole exposure in hematological pediatric patients could be low: the analysis of 16 patients showed that dosage adjustment for isavuconazole was required in 3.3% of cases (*p* < 0.001) and at the first real-time therapeutic drug monitoring, the reach of the desired target was higher for isavuconazole than voriconazole (83.3% vs. 10%, *p* = 0.008) [105].

Ashkenazi-Hoffnung et al. reported 4 cases of pediatric Mucormycosis (three who also had hematological malignancies, one with major trauma) treated with isavuconazole as salvage therapy: the 4 patients (included one with disseminated disease) showed complete clinical, radiologic and mycologic responses [109]. Review of the literature found a total of 8 other patients with hematological malignancies affected by Mucormycosis who showed similar response rates after isavuconazole therapy [109].

Decembrino et al. studied the use of isovuconazole as a treatment or prophylaxis for 29 patients with hematological malignancies (10 patients received only chemotherapy, 19 underwent HSCT) [110]. Overall, 5 patients received isavuconazole as prophylaxis and none of them developed IFD in the 90 days after the end of treatment. Of the 24 patients treated with isavuconazole as first-line or rescue therapy, 12 patients showed a complete response and 5 had a partial response (success rate of 70.8%). A total of six patients experienced adverse effects during treatment (elevation of transaminases, bilirubin or creatinine), but all were able to complete the treatment [110]. Zimmerman et al. had recently similar results in a cohort of 15 hemato-oncological patients (10 of these later also underwent HSCT) treated with isavuconazole: 9 received the drug as curative treatment, 6 as prophylaxis; no patient showed progressive fungal disease and 6 had complete response at day 90 [111]. No suspension of chemotherapy or immunosuppression was needed and no drug-to-drug interaction was noted.

Overall, these studies showed the efficacy of isavuconazole in the treatment of various IFD (even aspergillosis) in pediatric immunocompromised patients (including the setting of HSCT), in whom the extensive combination of chemotherapy and immunosuppression makes the use of antifungal drugs demanding.

### 9.2. Rezafungin

Rezafungin is a novel echinocandin with promising results in the latest trials [112]. Considered a “next-generation” echinocandin, it has demonstrated good efficacy in vivo against *Candida* spp. (even against MDR strains) and *Aspergillus* spp. [113,114]. Preclinical data showed no hepatotoxicity and minimal inhibition of CYP450 enzymes, thus outlining favorable clinical implementation; in a phase 1 study on healthy adults, no deaths, serious adverse events or withdrawals were recorded [115].

The phase 2 study STRIVE (multi-center, randomized, double blind) compared the administration of rezafungin vs. caspofungin with fluconazole stepdown in adult patients with candidemia and/or invasive candidemia: the treatment was generally well tolerated (no deaths, two serious adverse events related to rezafungin, fully reversible) and all primary safety and efficacy objectives were met [116].

Recently, Cidara Therapeutics published the results of the ReSTORE study, a phase 3 efficacy and safety study, in which rezafungin showed non-inferiority to caspofungin in the treatment of candidemia, meeting all primary endpoints [117]. Currently, Cidara Therapeutics is performing another phase 3 trial (ReSPECT; NCT04368559) with the aim of evaluating the use of rezafungin for the prophylaxis of fungal infections in adult patients undergoing HSCT [112,118].

Thanks to the encouraging results, the FDA has approved rezafungin as a Qualified Infectious Disease Product with fast-track status for its development program, both for the treatment and prophylaxis of candidemia in HSCT patients.

At the moment, no data regarding pediatric populations are available for the use of rezafungin, but hopefully, after the ReSTORE trial results, pediatric patients will be enrolled in order to evaluate the possibility of giving clinicians another therapeutic option.

### 9.3. Antimicrobial Peptides

In recent years, antimicrobial peptides (AMP), i.e., small agents representing the first line of defense in many living organisms [119], have generated interest in many researchers and new discoveries have been made regarding their potential use as therapeutic agents [120].

Nikkomycin Z is an AMP that was initially studied for the treatment of coccidioidomycosis [121]. While it has limited activity as monotherapy, it has shown the potential to enhance the activity of echinocandins, fluconazole and itraconazole in the combined treatment of *Candida* spp. and *Aspergillus* spp., thus reducing the development of resistance [122,123]. Nikkomycin Z completed a phase 1 clinical development trial with a good safety profile in healthy humans [124].

VL-2397 is an AMP that shows excellent in vitro activity against *Aspergillus* spp. (including azole-resistant strains) [125,126] and in vivo against invasive candidiasis in murine models [127]. It has completed a phase 1 study that has been well tolerated in healthy adults without reporting serious adverse effects [128]. A phase 2 trial for the treatment of aspergillosis in leukemia and HSCT patients was planned but unfortunately stopped for business decisions [129].

MAT2203 is a new form of encochleated amphotericin B, in which a solid phospholipid bilayer protects the molecule from degradation in the gastrointestinal tract, enabling oral administration and reducing toxicity [130]. A phase 1 study demonstrated that a single dose of MAT2203 was well tolerated [131] and a phase 2a clinical trial for the treatment of refractory mucocutaneous candidiasis is ongoing [132]. The FDA has approved the fast-track status for the treatment of invasive candidiasis, aspergillosis and the prevention of IFIs in patients undergoing immunosuppressive therapy [133].

Despite their potential to be safer than conventional drugs, more than a thousand antifungal AMPs have been described, but few of them have reached the market or at least advanced clinical stages [134,135], as in general, the design of clinical trials for novel antifungals is never easy [136]. The potential development of in silico peptide optimization, alongside the opportunity to design peptides *de novo* [137], will hopefully open a new era for the full exploitation of antifungal AMP capabilities.

## 10. Antifungal Stewardship

Antimicrobial stewardship is mostly focused on decreasing the rapid spread of antimicrobial resistance [138,139,140]. In recent years, the same need for antifungal stewardship has emerged among researchers [141]. In an interesting article, Ananda-Rajah et al. highlighted two main reasons for antifungal stewardship: the emergence of resistance to classical antifungal drugs and the high cost of antifungal treatments [142]. Antifungal stewardship includes the implementation of antifungal guidelines, interventions that can easily be translated from guidelines into everyday clinical practice, and careful post-prescription feedback [143,144].

Ferreras-Antolin et al. published the results of their surveys carried out weekly in 12 English hospitals during the course of 26 weeks regarding antifungal prescriptions in pediatric patients [145], most of whom were affected by cancer. Overall, 63.9% of patients received antifungal prophylaxis, and 40.2% of them were not considered high-risk patients. Antifungal treatment was needed for 36.1% of patients, and proven IFD was diagnosed in a total of 78 patients of the 656 registered in the surveys: candidiasis was the most prevalent and 12 children had invasive mold infection. The most prescribed drug, both for prophylaxis and treatment, was liposomal amphotericin B, representing 467 of the total 1227 antifungal prescriptions, followed by itraconazole with 265 prescriptions. The authors stated that the elevated number of antifungal prescriptions for non-high-risk patients and the large number of children with suspected IFD who received antifungal drugs were some of the issues enforcing the urgent need for antifungal stewardship and educational programs [145].

The first results on the implementation of antifungal stewardship in a pediatric hematology-oncology unit come from a study by Santiago-Garcia et al., in which a multidisciplinary team established and supervised a protocol for the management of IFD [146]. During the period covered by the study, 56 pediatric patients received antifungal agents for a total of 185 episodes (58.9% as prophylaxis, 34.6% as empiric treatment and 6.5% as targeted therapy). The authors evaluated the performance of prescribing pediatricians attending an educational course on IFD. The percentage of correct answers improved drastically after the course: 19.7% at the post-course assessment, 21.1% at the 6-month assessment and 16.6% at the final 12-month assessment.

In conclusion, antifungal stewardship can be considered a still emerging “branch” of antimicrobial stewardship and adequate training for physicians who take care of cancer patients is urgently needed [147,148,149].

## 11. Conclusions

Despite improvements in antileukemia therapy and supportive care, infections remain a major cause of morbidity and mortality in pediatric patients. IFDs represent a minority of cases in comparison to bacterial and viral infections, but their impact might be far more serious, especially when prolonged antifungal therapy or invasive surgical treatments are required to eradicate colonization. The latest update of the ECIL guidelines has provided a helpful tool to improve the prevention, diagnosis and treatment of IFDs. Nevertheless, a single approach could not be applied to all patients, since children often present various complications (especially in the case of HSCT) and often require tailored therapy. Moreover, while the *Aspergillu*s infection rate appears stable, the increase in Candidemia due to resistant species might hint that in the next few years therapeutic options should be reconsidered in light of new epidemiological data. Finally, as exciting as it might be, CAR-T treatment and other immunotherapies present a new challenge in a totally unprecedented new cohort of patients with unique needs and complications that still need to be extensively studied. Definitive guidelines on IFD treatment considering the evolving epidemiology of antifungal resistance, new therapeutic approaches in pediatric cancer, novel antifungal drugs and the importance of an appropriate antifungal stewardship are urgently needed.

## Figures and Tables

**Figure 1 antibiotics-11-01316-f001:**
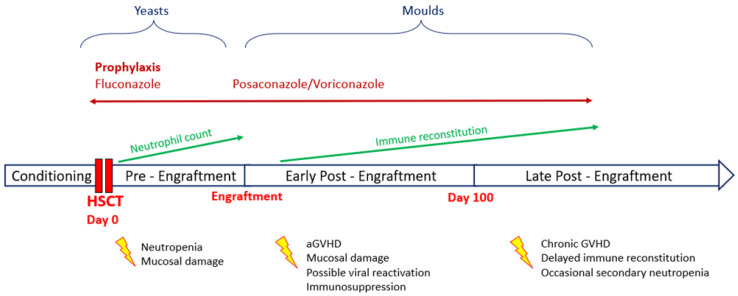
Timing, risk factors, epidemiology and prophylaxis indications for invasive fungal disease (IFD) in the context of hematopoietic stem cell transplantation (HSCT).

**Table 1 antibiotics-11-01316-t001:** Summary of main guidelines recommendations on antifungal prophylaxis in pediatric patients with cancer.

Guidelines	Recommendation	Strength of Recommendation
ECIL-8 [13]	Primary antifungal prophylaxis is strongly recommended for pediatric patients at high risk of IFD (acute myeloid leukemia, recurrent leukemia, high-risk acute lymphoblastic leukemia, allogeneic HSCT in the pre-engraftment and in the post-engraftment phase, augmented immunosuppressive treatment.	Strong
	Local epidemiology should be considered for the choice of the agent.	Strong
COG [39]	Antifungal prophylaxis must be administered in patients receiving treatment of acute myeloid leukemia that it is expected to result in profound and prolonged neutropenia, preferring a mold active agent.	Strong High-quality evidence
	Antifungal prophylaxis should be not administered to patients with cancer at low risk for IFD (pediatric patients with lymphomas and solid tumors), while it is routinary in children undergoing allogeneic HSCT pre-engraftment and in those receiving systemic immunosuppression for the treatment of GvHD.	Strong Moderate-quality evidence
	Do not administer antifungal prophylaxis to pediatric patients with acute lymphoblastic leukemia at low risk for IFD.	Strong Low-quality evidence
	Consider antifungal prophylaxis for pediatric patients with newly diagnosed and relapsed leukemia at high risk for IFD.	Weak Low-quality evidence

COG, Children’s’ Oncology Group; ECIL-8, 8th European Conference on Infections in Leukemia; HSCT, hematopoietic stem cell transplantation; GvHD, graft versus host disease; IFD, invasive fungal infection.

**Table 2 antibiotics-11-01316-t002:** Main studies on antifungal prophylaxis, empirical treatment and therapy of candidiasis, aspergillosis and mucormycosis.

Author	Design of Study	Results and Conclusions
Prophylaxis		
Fisher et al. [37]	517 children with AML randomly assigned to prophylaxis with caspofungin or fluconazole	23 total IFD. The 5-month cumulative incidence of IFD was 3.1% in the caspofungin arm vs. 7.2% in the fluconazole arm (*p* = 0.03). The study suggested that caspofungin might be considered for prophylaxis but was terminated early due to futility.
Dvorak et al. [38]	290 pediatric patients undergoing HSCT were randomized in two arms: caspofungin and triazole (fluconazole or voriconazole)	IFD cumulative incidence at day 42 was 1.4% in both groups (with no significative difference between fluconazole or voriconazole). Caspofungin did not significantly reduce the rate of IFD.
Empirical treatment		
Prentice et. al. [44]	100 adults and 204 children randomized in two groups, receiving either conventional amphotericin B, liposomial amphotericin B or liposomial amphotericin B at higher dose if pyrexia of unknown origin occurred	Liposomial amphotericin B at either 1 or 3 mg/kg was safer and possibly superior to conventional amphotericin B.
Maertens et al. [45]	Patients between 2 to 17 years of age with persistent fever and neutropenia were randomly assigned to receive caspofungin or liposomial amphotericin B.	Adverse drug related events were similar between the caspofungin and the amphotericin arms (clinical 48% [32.7–62] vs. 46.2% [26.6–66.6]). The two drugs resulted comparable in terms of tolerability, safety and efficacy.
Caselli et al. [46]	110 pediatric neutropenic patients with fever divided in two arms: high risk (receiving caspofungin or liposomial amphotericin B) and low risk (receiving caspofungin, liposomial amphotericin B or no drugs).	Complete response was achieved in 85.7% of high-risk patients (*p* = 0.72) and 87.5% of low-risk patients (with a rate of 87.5% alone in the no drug sub-arm, *p* = 0.41). Antifungal empirical therapy seemed unlikely to provide any benefit in low-risk patients.
Candidiasis		
Queiroz—Telles et al. [47]	106 pediatric patients treated for invasive candidiasis with micafungin or liposomial amphotericin B.	Treatment success observed for 35/48 (72.9%) patient cured with micafungin and 38/50 (76%) for the ones who received amphotericin B. The authors stated that the efficacy of the two drugs was similar.
Pappas et al. [48]	595 adult patients with candidemia or invasive candidemia were divided in three different groups: micafungin 100 mg/daily, micafungin 150 mg/daily, caspofungin 70 mg followed by 150 mg/daily.	Similar success rate (76.4, 71.4 and 72.3% respectively) were observed. Non-inferiority of micafungin to caspogfungin was proved.
Kullberg et al. [49]	450 adult patients affected by candidemia randomized to receive isovuconazole or caspofungin.	Successful overall response in the isovuconazole arm was 60.3% vs. 71.1% in the caspofungin arm. Non-inferiority of isovuconazole was not demonstrated.
Aspergillosis		
Maertens et al. [50]	527 adult patients with suspected mold disease randomized in two groups (isovuconazole vs. voriconazole)	All-cause mortality at day 42 from first drug dose was 19% with isovuconazole and 20% with voriconazole. Drug-related adverse events were fewer in the isovuconazole arm (42% vs. 60%, *p* < 0.001). Isovuconazole resulted in non-inferior to voriconazole.
Cornely et al. [51]	195 adult patients and 6 pediatric patients with confirmed mold infection received liposomial amphotericin B at 3 mg/kg/day or 10 mg/kg/day	Good response was achieved in 50% and of patients in the 3 and 10 mg/kg/day respectively (*p* > 0.05). A significant rate of nephrotoxicity and hypokalemia was observed in the higher-dose group. The 3 mg/kg/day dosage was effective and well tolerated as first-line therapy
Mucormycosis		
Lanternier et al. [52]	40 patients with proven or probable Mucormycosis (including two children) were scheduled to receive liposomial amphotericin B for one month before surgery.	Response rate at week 4 was 36% and increased to 45% at week 12, showing potential efficacy for a combined amphotericin/surgical treatment
Skiada et al. [53]	Retrospective analysis of 230 cases of zygomycosis.	Factors associated with survivability were history of trauma (*p* = 0.019), treatment with amphotericin B (*p* = 0.006) and surgery (*p* < 0.001), meaning that association of amphotericin administration and surgical treatment (if feasible) can improve survival.

AML, acute myeloid leukemia; HSCT, hematopoietic stem cell transplantation; IFD, invasive fungal infection.

**Table 3 antibiotics-11-01316-t003:** Main studies on the use of granulocyte transfusion in pediatric cohorts.

Author	Design of Study	Results and Conclusions
Sachs et al. [77]	27 children with hematologic disorder or malignancy and severe neutropenia with clinically and/or microbiologically documented severe infection (including 6 invasive aspergillosis and one disseminated candidemia) unresponsive to standard treatment received GTX.	25 out 27 patients cleared the infection, including the 6 children with aspergillosis, a great success rate, probably influenced by the low number of IFD considered and earlier start of GTX.
Pham et al. [78]	Retrospective observational analysis on GTX from stimulated and un-stimulated donors administered in pediatric HSCT patients in a single center from 2005 to 2010. In 19% of the cases, 153 GTXs were administered for IFD.	Most patients cleared the index infection, only one affected by candidemia did not. Survival between patients receiving GTX from stimulated and un-stimulated donors was not significantly different (*p* = 0.42). The retrospective nature of this study strongly limits its results.
Diaz et al. [79]	Retrospective review of 18 children with neutropenia or granulocyte disfunction receiving GTX.	13 patients had complete or partial response (two infections caused by *Fusarium* and *Histoplasma* spp. progressed). While the clinical benefit was evident, the retrospective nature and lack of a comparison group do not allow us to demonstrate the superiority of GTX alone against antimicrobials.
Nikolajeva et al. [80]	Retrospective analysis on 28 pediatric patients undergoing HSCT and receiving GTX (14 of them affected by proven, probable or possible IFD).	11 of the 14 patients with IFD survived, only one died for IFD progression. Interestingly, a low rate of GVHD was observed, but these results must be confirmed in larger cohorts.
Koc et al. [81]	Retrospective review on 9 pediatric hematology and oncology patients receiving GTX.	Clinical response rates after GTX was 90.9%, while mortality rate was 9%. The large limitations of this study are the small cohort considered and the absence of IFD.

GTX, granulocyte transfusion; HSCT, hematopoietic stem cell transplantation; IFD, invasive fungal disease.

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
