# Peer review of "Novel Insights into Fungal Infections Prophylaxis and Treatment in Pediatric Patients with Cancer"

_antibiotics, 2022, doi:10.3390/antibiotics11101316_

Round 1

Reviewer 1 Report

The manuscript by Ganluca Bossù et al. aims in the characterization of fungal infections prophylaxis and treatment in pediatric patients with cancer. The review provide an interesting added value in our comprehension what are the major problems that the scientific and clinical community faces with invasive fungal infections as complications in the treatment of certain forms of cancer in children. Thus I think that it justifies the publication of this manuscript in the Pathogens,  but not Antibiotics.  The paper is very well written, the tables are a valuable summary of some issues.

The pediatric population is defined in the European Union (EU) as that group of the population between birth and 18 years. On the other hands, the pediatric population encompasses several subsets and in accordance with current guidelines, the applied age classification of pediatric patients is: preterm newborn neonates; term and post-term neonates; infants (or toddlers): from 1 month (28 days) to 23 months; children: from 2 years to 11 years; and adolescents: from 12 years to less than 18 years. Could you please respond to the question whether the age range matters in respect of your summarizing?

Some corrections must be introduced before considering the paper for publication.  First of all, it is not clear why does data extrapolated from adult cohorts are presented in Table 2 while the review aims in characterization of pediatric patients.

Specific comments/remarks::

1.       Line 100 – please change “too have.” to “to have”

2.       All the text – “in vitro”, “in vivo”, “in silico”, “de novo”  - please use italics.

3.       Line 166  - beginning of a sentence with a capital letter

4.       Table 2, page 6 – “Good response was achieved in 50% and of patients in the 3 and 0 mg/kg/day respectively” – it should be “Good response was achieved in 50% and of patients in the 3 and 10 mg/kg/day respectively”.

5.       Line 362 – “More recent data106 suggested that...” – it should be: “More recent data [106] suggested that…

Author Response

The manuscript by Ganluca Bossù et al. aims in the characterization of fungal infections prophylaxis and treatment in pediatric patients with cancer. The review provide an interesting added value in our comprehension what are the major problems that the scientific and clinical community faces with invasive fungal infections as complications in the treatment of certain forms of cancer in children. The paper is very well written, the tables are a valuable summary of some issues.

Re: Thank you for your positive evaluation. We revised the manuscript according to your suggestions.

The pediatric population is defined in the European Union (EU) as that group of the population between birth and 18 years. On the other hands, the pediatric population encompasses several subsets and in accordance with current guidelines, the applied age classification of pediatric patients is: preterm newborn neonates; term and post-term neonates; infants (or toddlers): from 1 month (28 days) to 23 months; children: from 2 years to 11 years; and adolescents: from 12 years to less than 18 years. Could you please respond to the question whether the age range matters in respect of your summarizing?

Re: Age range has been clarified (p. 2).

Some corrections must be introduced before considering the paper for publication.  First of all, it is not clear why does data extrapolated from adult cohorts are presented in Table 2 while the review aims in characterization of pediatric patients.

Re: Clarified (p. 2).

Specific comments/remarks::

  1. Line 100 – please change “too have.” to “to have”
  2. All the text – “in vitro”, “in vivo”, “in silico”, “de novo” - please use italics.
  3. Line 166 - beginning of a sentence with a capital letter
  4. Table 2, page 6 – “Good response was achieved in 50% and of patients in the 3 and 0 mg/kg/day respectively” – it should be “Good response was achieved in 50% and of patients in the 3 and 10 mg/kg/day respectively”.
  5. Line 362 – “More recent data106 suggested that...” – it should be: “More recent data [106] suggested that…”

Re: All specific comments/remarks have been revised (pp. 3, 4, 6 and 10).

Reviewer 2 Report

Comments to the authors:

1.     The sentences in Abstract (lines 19-28) seem to be identical to the Conclusion (lines 484-494).

2.     ‘Fungal infecions’ (line 265), it should be ‘Fungal infections’

3.     The authors mentioned invasive fungal disease (IFD) treatment in pediatric patients treated with CAR-T (lines 265-304). The authors should give a definition of CAR-T and how CAR-T therapy works.

4.     Figure 1 should be made easy to understand, and the font size should be increased to make it easy to read.

Author Response

Comments to the authors:

Re: Thank you for the positive evaluation of our manuscript. We revised the text according to your suggestions.

  1. The sentences in Abstract (lines 19-28) seem to be identical to the Conclusion (lines 484-494).

Re: Abstract has been revised (p. 1).

  1. ‘Fungal infecions’ (line 265), it should be ‘Fungal infections’

Re: Corrected (p. 8).

  1. The authors mentioned invasive fungal disease (IFD) treatment in pediatric patients treated with CAR-T (lines 265-304). The authors should give a definition of CAR-T and how CAR-T therapy works.

Re: Clarified with a reference added (pp. 9 and 17).

  1. Figure 1 should be made easy to understand, and the font size should be increased to make it easy to read.

Re: Revised as recommended.